# Perceived Benefits and Forest Tourists Consumption Intention: Environmental Protection Attitude and Resource Utilization Attitude as Mediators

Bin Zhou [1], Siyi Liu [1], Hu Yu [2,*], Dongfang Zhu [2] and Qihao Xiong [1]

1   Marine Economics Research Center, Donghai Academy, Department of Tourism, Ningbo University, Ningbo 315211, China; njuzhoubin@163.com (B.Z.); lsyyy64111@163.com (S.L.); banaisen@163.com (Q.X.)
2   Institute of Geographic Sciences and Natural Resources, CAS, Beijing 100101, China; zdfashd@163.com
*   Correspondence: yuhu@igsnrr.ac.cn

**Abstract:** Establishing a relationship model between environmental protection and resource utilization attitude and consumption intention is the key to promoting the sustainable development of forest tourism. From the standpoint of the Stimulus–Organism–Response (SOR) framework, the purpose of this study is to explore the complex causal relationships between perceived benefits, attitudes toward environmental protection, resource utilization attitudes, and consumption intentions in the context of forest tourism. The research data have been collected using a questionnaire survey of 436 tourists at Siming Mountain in the suburbs of Ningbo city, China. Furthermore, it is analyzed by structural equation modeling. The results indicate a positive correlation between the perceived benefits and tourists' consumption intention that is mediated by the tourists' attitude toward resource utilization. Although the independent mediating effect of environmental protection attitude is not supported in this study, both attitudes have played a chain-mediating role between perceived benefit and consumption intention. This study contributes to the existing knowledge by measuring the impact of perceived benefits and environmental attitudes of forest tourists on consumption intentions.

**Keywords:** forest tourism; perceived benefits; consumption intention; environmental protection attitude; resource utilization attitude; Siming Mountain

## 1. Introduction

With the continuous expansion of health awareness, people are now more eager to maintain or improve their physical and mental health through tourism [1]. Reducing unpleasant environmental and social impacts, increasing economic impact, and encouraging meaningful experiences, forest tourism is growing rapidly in the tourism industry [2]. For instance, wellness tourism may meet various sorts of tourists' health needs such as body, mind, spirit, and environmental [3]. Indeed, this is gradually becoming a marked trend [4–8]. The wellness tourism industry is booming worldwide and raising a niche market in the tourism field, with market sales expected to reach USD 919 billion by 2022 [9].

Studies have pointed out that the willingness of people in five Asian countries, including Indonesia, Japan, South Korea, etc., to return to nature has increased in the context of the COVID-19 global pandemic [10]. Forest tourism, as a type of nature tourism, is beneficial to the body and mind, and also highly accessible. Particularly, it has been noted that walking on forest park trails brings significantly more benefits to human vascular function than in urban parks [11]. Additionally, empirical studies have found that forest environments have not only physiological health benefits such as lowering human blood pressure [12], fatigue reduction [13], enhancing immunity [14], reducing inflammation [15], attributes of adjuvant anti-cancer therapy [16], etc., but also good psychological effects on humans, such as reducing visitor stress and relieving anxiety [12], alleviation of depression [17],

revitalization [18], etc. In addition, forest therapy, an activity closely related to forest tourism, which is an activity to restore the body and mind and improve physical fitness in the forest, has been reported to be associated with a reduction in anxiety, positive emotional states, vitality, friendship and positive thinking [19,20]. In Korea, the number of tourists participating in forest healing has increased rapidly in recent years [21]. It can be seen that a pleasant ecological environment improves both physical and mental health [22,23]. Forests containing abundant natural landscapes serve as the context of forest ecosystem services and extend important resources to enhance the well-being of the people [24,25].

China is wealthy in terms of forest resources, with a forest area of 220 million hectares in 2020, a forest coverage rate of 23.04%, and a forest stock above 17.5 billion m$^3$ [26]. There is a rapid increase in forest tourism in China. National forest parks are an integral part of China's nature reserve system. It serves as an important position to popularize nature knowledge and spread the concept of ecological civilization, as well as an important carrier of forest ecotourism. According to the State Forestry and Grassland Administration, there are 3564 forest parks at all levels in China (including 897 national forest parks, 1459 provincial-level forest parks, and 1208 county-level forest parks). Moreover, the number of national forest tourism tourists reached 1.8 billion in 2019 [27]. Furthermore, as the main site for the development of ecotourism, mountain tourism, recreation, and leisure tourism in China, the tourism revenue of national forest parks has exceeded a trillion yuan by the end of 2019 [28]. Chinese forestry managers employ the concept of ecotourism to guide the development of forest tourism, i.e., under the premise of protecting nature. This institution scientifically and rationally utilizes forest resources to fulfill people's desires for health and recreation [29]. In China, the conjoining of forest wellness, traditional Chinese medicine, and physical therapy culture has become an ideal development model for several businesses to use in pursuing product upgrades. It also gives the tourists a superior health experience, and therefore it has gradually developed into a new industry.

Consumers' perceived benefits of products or services serve as the primary factor to influence their consumption intentions [30–32]. Tourists' perceptions regarding a destination's benefits play a decisive role in shaping their attitudes and subsequent behavioral changes [33,34]. Positive perception encourages tourists' consumption of the tourism products and is also a key factor in determining the tourists' willingness to revisit a specific tourist destination [35–37]. Moreover, attitude is one of the critical factors affecting the consumption intentions of consumers [38,39]. Since tourism activities that are located in the natural environments are unique, they allow for activities that fulfill tourists' demand for a natural experience in a non-exclusive and non-competitive manner [40]. Consequently, for natural tourism activities, tourists' environmental attitudes act as a vital factor that influences the consumption intentions of the tourists [40–43].

Kellert developed and applied a set of value measures to subsequent studies, on the basis of which the *biophilia hypothesis* was proposed [44–47]. Some research studies advocate that the tourists' environmental attitudes serve as the core concept of the individual value to protect and improve the environment [48–50]. However, some environmental psychologists claim that environmental attitude is a two-pronged concept, which should be categorized into attitudes toward environmental protection, on the one hand, and attitudes toward appreciation and utilization of nature, on the other [51,52]. While the former is a selfless and altruistic force, the pure appreciation of nature and utilization of nature for personal recovery, recreation, and development are more self-interested practices [52,53]. Notably, both of these are relatively independent but closely related factors [51], and both are significant in affecting consumer preferences [54].

The Stimulus–Organism–Response (SOR) framework is a framework to explain the external stimulus that affects an individual's behavior [55]. An individual's internal emotional response is the mediating variable that ascertains the external response through this framework. The SOR model is frequently carried out to elucidate the correlation between the consumption environment and consumption behavior [56–58]. Besides, it has been applied in tourism studies in recent years [59,60]. Based on the SOR framework

and taking the Siming Mountain in the outskirts of Ningbo City, Zhejiang Province, as a case site, this study aims to investigate the impact of forest tourist perceived benefits on consumption behavior. In addition to this, it analyzes the mediating role of tourists' attitudes toward environmental protection and resource utilization. In short, the influence of forest tourists' perceived benefits on consumption intention is the focus of this research study as it provides a theoretical basis for the development and practice of forest tourism in the Siming Mountains.

### 1.1. Perceived Benefit and Consumption Intention

Perceived benefit is the overall assessment of product utility [61]. It is defined as the perceived likelihood of the positive consequences of consumer behaviors [32,62]. Tourists' perceived benefits are determined by the actual contact of tourists with destinations as well as the interaction between the two. These perceived benefits increase with the augmentation of tourists' sensory experiences [63]. Forests serve as a vital natural environment where visitors can experience a variety of activities, such as hiking, hot springs, camping, fishing, forest education, and viewing rare animals and plants [64]. Therefore, forest tourism offers a wide range of benefits including recreation, health, medical treatment, culture, and education [65–69]. Furthermore, the pursuit of wellness is a major motive of forest tourists [3,70]. With pure air, high-quality surface water, dense anions, a high concentration of terpenes, and a pleasant climate, forests enable tourists to enjoy physical and mental health benefits [71,72]. In addition, the quality and value of products and services are considered perceived-value benefits [73,74].

Perceived benefit, as the sum of product advantages that fulfill consumers' needs and desires [75], plays a key role in motivating consumption intention and consumption behaviors [32,76]. Priem [31] emphasized the maximization of consumers' perceived benefits as a means to gain a competitive advantage. In the tourism literature, some existing studies have also confirmed the impact of tourists' perceived benefits (from the destination) on the expenditure on tourism projects. Henderson-Wilson et al. [77] highlighted that urban park visitors show a willingness to spend more, provided these visitors believe that they could gain more benefits from exercise, social interaction, and relaxation experience. Park and Song [78] divided tourists into three categories based on the perceived benefit preferences for urban lake parks, and their consumption intentions for parks were influenced by the categories of tourists.

### 1.2. Perceived Benefit, Environmental Protection Attitude, and Consumption Intention

Natural tourism destinations not only provide the tourists with opportunities to experience nature but also potentially increase their understanding of the environment, deepen their connection with nature, and account for the formation and change of the tourists' environmental attitudes [79]. The natural environment of an ecotourism destination also has an educational function [80]. Tourists could increase environmental awareness through ecotourism experience and pay more attention to the protection of resources and the environment [81]. The research on various natural tourist destinations, such as national parks, protected areas, and forests, has demonstrated that tourists' perceived benefits regarding the natural environment positively impact their environmental protection attitude [82–84].

Some researchers believe that the environmental protection attitude is the most influential predictor of green consumption behavior [85]. Tourists with a strong environmental awareness prefer to participate in ecotourism [40,41,86], stay in green hotels [87,88], and purchase environmentally friendly products [89]. Tourists with an environmental protection attitude are more willing to provide financial support for the management of eco-scenic spots in order to reduce the negative impact of tourism and promote its sustainable development [90]. It is also worth mentioning here that tourism development can destroy the habitat of precious wildlife, therefore, tourists show a willingness to pay specifically for the conservation fees in order to protect rare wild animals and plants [42,43].

A nature-based tourist behavior model has been proposed and proven [91,92]. This model suggests that the leisure and entertainment experience in the natural environment can improve tourists' impression related to the biosphere, enhance their attitude toward environmental protection, and promote their adoption of environmentally responsible behaviors. Notably, the environmental attitude plays a mediating role between tourist experience and environmentally responsible behavior. Green consumption behavior is regarded as a type of environmentally responsible behavior. Considering that environmental protection attitude has an important driving effect on the green consumption behavior [85], it can be inferred that environmental protection attitude also plays a mediating role between the tourists' perceived benefits and consumption intention.

### 1.3. Perceived Benefit, Resource Utilization Attitude, and Consumption Intention

Perceived benefits contribute to positive attitudes toward specific tourist destinations [93]. Evidence suggests that in natural tourist destinations, the direct experience of the natural environment increases the tourists' appreciation of the environment [84]. In addition, some studies advocate that the higher the tourists' willingness to establish resources at their destinations, the stronger is their willingness to experience tourism activities and purchase tourism commodities [94]. Thus, tourists' positive perceptions related to the destinations stimulate their attitudes toward using destination resources, thereby promoting tourists' willingness to purchase tourism products developed by using local resources. According to the SOR stimulus framework, attitude commonly plays a mediating role in stimulus and response, which is supported by Zhao and An [95], who found that the forest tourists' positive attitudes toward destination resources exert a significant mediating effect between the perceived benefits and tourist behaviors. Thus, the tourists' resource utilization attitude acts as an intermediary in the psychological pathway of perceived benefits to the consumption intention.

### 1.4. Chain-Mediating Effect of Environmental Protection Attitude and Resource Utilization Attitude

Environmental protection attitude and resource utilization attitude are two aspects of environmental attitude [52], which are produced simultaneously and influence tourists' behaviors when tourists are in natural destinations [53]. This premise has been suggested by many empirical studies. For example, Russell and Russell [84] reported that the experience of tourists in national parks not only enhances their awareness of environmental protection but also their appreciation of the value of the ecological environment. In addition, it stimulates tourists to financially support the park. Some research studies have investigated the tourists' support related to the conservation and management of natural scenic spots and reported that tourists were more willing to spend money on biodiversity conservation after their tourism activities [96,97]. Furthermore, Kaiser et al.'s [51] quantitative research established that environmental protection attitudes were significantly correlated with the appreciation and utilization of nature.

With the concept of harmonious coexistence between humans and nature becoming part of mainstream ideology, certain studies consistently propose that the sustainable development of tourism destinations is a crucial method to ensure environmental protection [98]. Under the influence of this concept, tourists' attitudes and behaviors toward the natural environment have also evolved over time. In addition, several empirical studies have highlighted that people are willing to pay higher prices for more environmentally friendly and sustainable tourism [86]. However, few studies have explored the joint effect of environmental protection attitude and resource utilization attitude on tourism consumption intention. Commonly, researchers claim that environmental protection attitude is an emotional driving factor for environmental behavior [91,92]; however, Kaiser et al. [51] highlighted that ecological behavior could also stem from a self-interested attitude grounded in the personal benefits of nature experiences. Not only in natural tourism destinations, Zhang and Chen [94] also reported that tourism development attitude plays a mediating

role between tourists' protection attitude toward intangible cultural heritage and tourism consumption intention. Thus, tourists' perceptions of nature can awaken their awareness of environmental protection. Subsequently, environmental protection awareness can promote tourists' support for sustainable development of resources, thereby enhancing the tourists' consumption intention.

## 2. Materials and Methods

### 2.1. Research Hypotheses

This study has assessed the relationship between tourists' perceived benefits and consumption intentions in Siming Mountain National Forest Park. Additionally, based on the previous literature review and discussion, this study sought to explore the multiple mediating roles of tourists' attitudes toward environmental protection and resource utilization between perceived benefits and consumption intentions. The final theoretical model is shown in Figure 1. Hence, the following hypotheses are proposed in this study:

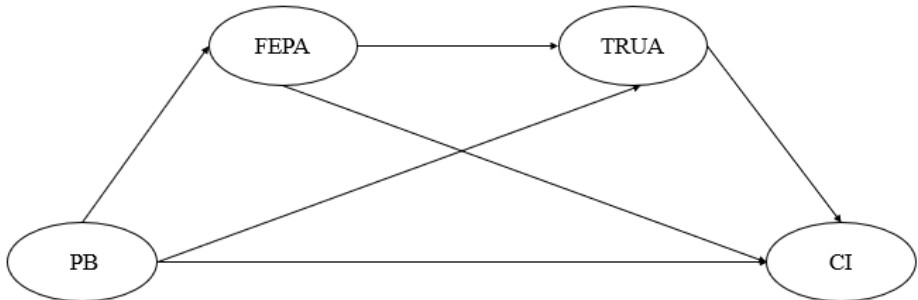

**Figure 1.** The theoretical model in this paper. Note: PB = Perceived Benefit; FEPA = Forest Environmental Protection Attitude; TRUA = Tourism Resources Utilization Attitude; CI = Consumption Intention.

**Hypothesis 1:** *Forest tourists' perceived benefits exert a positive impact on consumption intention.*

**Hypothesis 2:** *Tourists' perceived benefit exerts a positive impact on attitude toward forest environmental protection.*

**Hypothesis 3:** *Forest environmental protection attitude exerts a positive impact on tourists' consumption intention.*

**Hypothesis 4:** *Forest environmental protection attitude plays a mediating role in the correlation between perceived benefits and consumption intentions of tourists in forest tourism.*

**Hypothesis 5:** *Perceived benefits of forest tourism exert a positive impact on tourists' attitudes toward the utilization of forest resources.*

**Hypothesis 6:** *Attitudes toward utilization of forest resources exert a positive impact on tourists' consumption intention.*

**Hypothesis 7:** *Forest resource utilization attitudes play a mediating role in the correlation between tourists' perceived benefits and consumption intention.*

**Hypothesis 8:** *Forest environmental protection attitude exerts a positive impact on tourist resource utilization attitude.*

**Hypothesis 9:** *Forest environmental protection attitude and resource utilization attitude play a chain-mediating role between tourists' perceived benefits and consumption intention.*

### 2.2. Study Site

The Siming Mountain spans over five counties—Shengzhou and Shangyu in Shaoxing; Yuyao, Haishu, and Fenghua in Ningbo—with an altitude of 600–900 m. With a subtropical monsoon climate, the Siming Mountain has four distinct seasons and mountain climate characteristics. The climate is mild and humid, with an average annual temperature of 13 °C. It has a total area of 6665 hectares, with a standing wood accumulation of 350,000 cubic meters and a forest coverage rate of over 96%. Most of the existing vegetation on the mountain is artificial and natural secondary forest, and the vegetation types are primarily evergreen broad-leaved forests and evergreen deciduous broad-leaved mixed forests. Siming Mountain National Forest Park also includes various scenic spots (Figure 2).

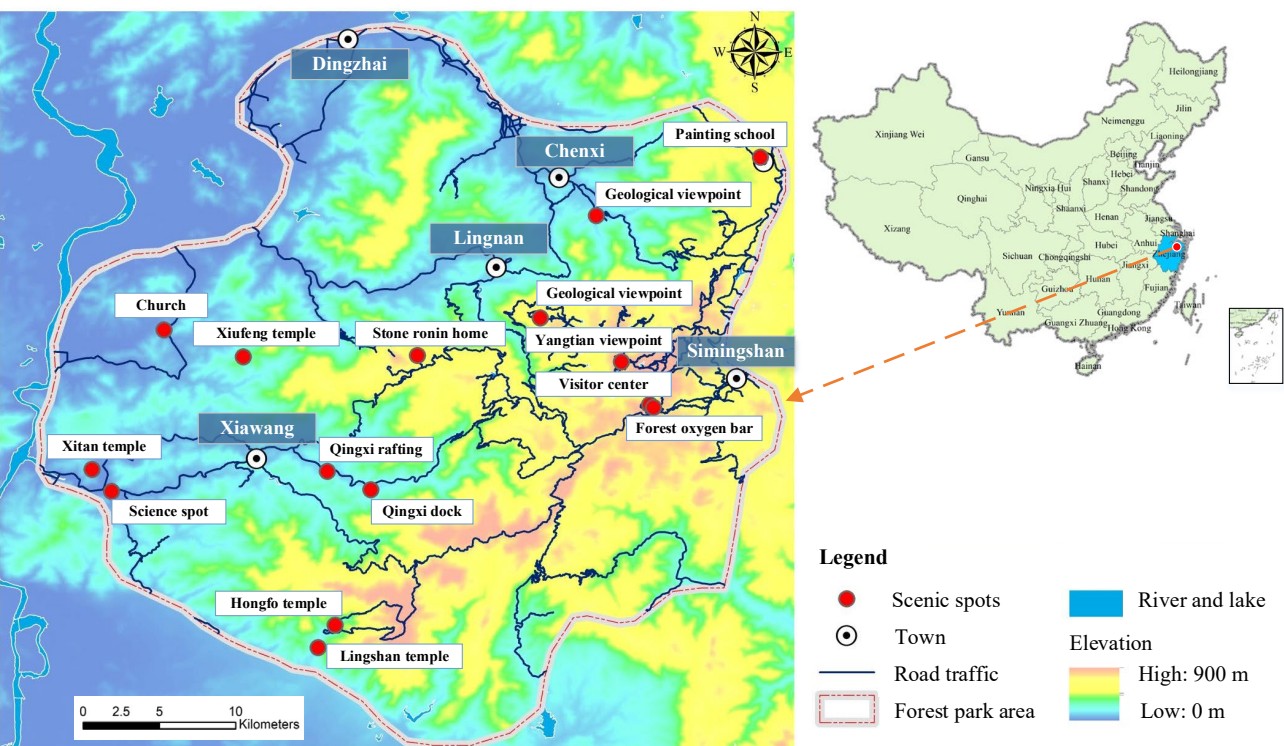

**Figure 2.** The Map of the Siming Mountain.

### 2.3. Measures

Following the post-positivist paradigm, a self-reported questionnaire was designed for this study, based on measures from the previous literature. The first part of the questionnaire was the survey of tourists' behaviors, which primarily included the times of forest tourism experience in Siming Mountain, duration of stay, transportation, travel companions, travel purpose, and consumption amount. The second part featured the measurement of four study variables: tourists' perceived benefit, attitude toward environmental protection, attitude toward tourism resources utilization, and consumption intention. The items used for measuring tourists' perceived benefits of forest tourism were based on the studies of Chen et al. [99], Lee et al. [100], and Ohe et al. [23]. The questions were used to measure the environmental protection attitude and tourism resources utilization attitude which were derived from Milfont and Duckitt [52] and Bogner & Wiseman [101], respectively. The measure of consumption intention included four items which were adapted from Yadav and Pathak [102]. These items were modified in the context of forest tourism, and a scale of 26 items (shown in Appendix A) was finally adopted in this study. This study used the seven-point Likert scale method to quantify each item. The third part consisted of tourists' demographic characteristics and included content such as gender, education, age, occupation, marital status, and annual family income level.

*2.4. Data Collection*

This research survey was conducted between 22 March 2018, and 3 May 2018. The survey sites were the Siming Mountain National Forest Park, Danshan Chishui Scenic Area, and the hiking trails in the Siming Mountain area. The principle of convenience sampling was followed; the researchers selected one visitor from every five visitors who passed by and invited the respondents to finish the questionnaire with their consent, during which, if the respondents did not understand some items on the scale, the researchers would explain such items to them. To ensure the authenticity of the questionnaire, the researcher also ensured to collect the questionnaire on the spot. Lastly, each respondent was given a small gift prepared by the researchers. A total of 505 questionnaires were distributed in this study, of which 436 were valid questionnaires, with an effective rate of 86.34%.

The following data processing procedure was performed after the collection of the data. Firstly, the reliability and validity of the scale were tested. Secondly, we administered the exploratory factor analysis (EFA) and confirmatory factor analysis (CFA). Afterward, the structural equation modeling (SEM) was constructed to examine the model's fitness and to verify the validity of the research hypotheses. Finally, the Bootstrapping method was used to test the mediating effect. The measurement of mediating effects consisted of three parts: the analysis of specific mediating effects, the analysis of total mediating effects, and the comparative analysis of specific mediating effects. All data analysis was performed using SPSS 25.0 (IBM Corporation, Armonk, NY, USA) and AMOS 24.0 (IBM Corporation, Armonk, NY, USA).

## 3. Results

*3.1. Respondent Profile*

In this study, the proportion of female tourists (56%) is higher than that of men (44%). The proportion of tourists staying for 1 and 2 days was 45.9% and 46.1%, respectively. Self-driving is the dominant mode of transportation for tourists (60.1%), and most tourists traveled with their families (45.4%). In terms of age, tourists aged 26–35 represent the largest group, accounting for 44%. The educational background of the sample is primarily undergraduates (39.4%), followed by junior high school and below (17.2%). The proportion of married tourists (60.6%) is significantly higher than the unmarried tourists (39.4%), Moreover, 68.8% of tourists have an average monthly family income, while 21.6% have an above-average income. Regarding tourist occupations, employees of private enterprises, students, and employees of state-owned enterprises account for a relatively large proportion, 18.6%, 17.7%, and 16.1%, respectively.

*3.2. EFA*

The reliability test results reveal that the Cronbach scale value is 0.911 which indicates that the scale has good reliability and stability. The validity test results reveal that the KMO is 0.925, which is higher than the general standard of 0.7, and the Bartlett sphericity test is significant ($p < 0.001$). In EFA, when the factor loading of an item is <0.5 or when the factor loading of an item on two or more principal factors is greater than 0.5, the question item should be deleted. Finally, 24 items are retained to form six common factors (Table 1), and the cumulative variance contribution rate is 67.738%. Per the meanings of the items, the six common factors are as follows: perceived functional benefit, perceived value benefit, perceived health benefit, attitude towards forest environment protection, tourism resources utilization attitude, and consumption intention. To simplify the model and facilitate the estimation of parameters, perceived functional benefit, perceived value benefit, and perceived health benefit are aggregated into a second-order variable "perceived benefit of forest tourist".

**Table 1.** Results of exploratory factor and confirmatory factor analysis.

| Variable/Construct | MEAN | SFL EFA | SD | Cronbach' α | SFL CFA | CR (AVE) |
|---|---|---|---|---|---|---|
| **Perceived benefit of forest tourist** | | | | | | |
| **Perceived functional benefit** | | | | | | |
| The forest tourism of Siming Mountain have recreational value | 5.600 | 0.690 | 0.911 | | 0.680 | |
| The forest tourism of Siming Mountain have health preservation value | 5.500 | 0.772 | 0.896 | 0.805 | 0.742 | 0.807 (0.511) |
| The forest tourism of Siming Mountain have medical value | 5.530 | 0.715 | 0.856 | | 0.720 | |
| The forest tourism of Siming Mountain have cultural value | 5.610 | 0.765 | 0.987 | | 0.716 | |
| **Perceived value benefit** | | | | | | |
| Siming Mountain forest tourism has a distinct theme | 4.230 | 0.806 | 1.113 | | 0.788 | |
| Siming Mountain forest tourism has a reasonable consumption level | 4.610 | 0.715 | 1.116 | 0.808 | 0.654 | 0.814 (0.524) |
| The price of Siming Mountain forest tourism is moderate | 4.850 | 0.611 | 1.117 | | 0.648 | |
| Siming Mountain forest tourism provides professional services | 4.130 | 0.792 | 1.081 | | 0.792 | |
| **Perceived health benefit** | | | | | | |
| Siming Mountain forest tourism can relieve pressure | 5.270 | 0.726 | 1.051 | | 0.715 | |
| Siming Mountain forest tourism makes people happy | 5.130 | 0.726 | 0.974 | 0.828 | 0.778 | 0.830 (0.550) |
| Siming Mountain forest tourism can keep healthy | 5.220 | 0.799 | 1.091 | | 0.759 | |
| Siming Mountain forest tourism can enhance physical fitness | 5.030 | 0.710 | 1.116 | | 0.713 | |
| **Environmental protection attitude** | | | | | | |
| I think it's very important to protect the forest tourism resources and environment in Siming Mountain | 5.450 | 0.714 | 1.102 | | 0.739 | |
| I'd like to participate in the forest tourism resources and environment protection in Siming Mountain | 5.280 | 0.769 | 1.045 | 0.771 | 0.700 | 0.772 (0.530) |
| I'd like to prevent the destruction of forest tourism resources and the environment in Siming Mountain | 5.330 | 0.767 | 1.143 | | 0.745 | |
| **Tourism resources utilization attitude** | | | | | | |
| It is necessary to develop forest tourism in Siming Mountain | 5.280 | 0.701 | 0.984 | | 0.772 | |
| The development of forest tourism in Siming Mountain should reflect local characteristics | 5.240 | 0.703 | 0.999 | | 0.780 | |
| The development of forest tourism in Siming Mountain should reflect the health-preserving culture | 5.260 | 0.735 | 0.980 | 0.869 | 0.738 | 0.870 (0.572) |
| Willing to see more Siming Mountain forest health project | 5.400 | 0.775 | 0.908 | | 0.749 | |
| Willing to experience the new Siming Mountain forest tourism project | 5.320 | 0.714 | 0.924 | | 0.740 | |
| **Consumption intention** | | | | | | |
| I would like to consume the forest tourism in Siming Mountain | 5.360 | 0.756 | 1.278 | | 0.679 | |
| I would like to consume the forest tourism products in Siming Mountain | 5.120 | 0.773 | 1.356 | 0.885 | 0.842 | 0.887 (0.666) |
| I would like to experience the forest culture of Siming Mountain | 5.100 | 0.755 | 1.310 | | 0.840 | |
| I would like to consume receiving a forest education in Siming Mountain | 5.230 | 0.800 | 1.328 | | 0.887 | |

Note: SFL = standardized factor loading; SD = Standardized error; CR = composite reliability; AVE = average variance extracted.

### 3.3. CFA

The standardized factor loading values of the first-order variables and the observed variables is 0.648–0.887 (Table 1), all of which met the standard of 0.5. In addition, the factor loading values of the second-order variable "perceived benefit of forest tourist" and the first-order variable perceived functional benefit, perceived value benefit, and perceived health benefit are 0.739, 0.715, and 0.777, respectively. This demonstrates that the first-order variable has a strong explanatory power over the second-order variable. Table 2 shows the convergence validity and discriminant validity test results of the model. The standardized factor loading of each item is >0.5, the average variance extracted (AVE) is greater than 0.5, and the combined reliability is >0.7, which reaches the ideal standard of convergence validity. This suggests that each item is representative of the corresponding variables, and the model of CFA is substantially reliable. Moreover, the diagonal values in the table are the square root of the AVE of each latent variable, and all are higher than 0.70. The comparative analysis reveals that the square root of the AVE of each variable is higher than the correlation coefficient between constructs which highlights that each variable has good discriminative validity.

**Table 2.** Results of convergence and discriminative validity test.

| Latent Variables | 1 | 2 | 3 | 4 | 5 | 6 |
|---|---|---|---|---|---|---|
| PB1-Perceived functional benefit | 0.715 | | | | | |
| PB2-Perceived value benefit | 0.424 ** | 0.724 | | | | |
| PB3-Perceived health benefit | 0.457 ** | 0.503 ** | 0.742 | | | |
| Environmental protection attitude | 0.325 ** | 0.361 ** | 0.427 ** | 0.728 | | |
| Tourism resources utilization attitude | 0.466 ** | 0.438 ** | 0.452 ** | 0.558 ** | 0.756 | |
| Consumption intention | 0.464 ** | 0.423 ** | 0.456 ** | 0.471 ** | 0.585 ** | 0.816 |

Note: ** $p < 0.01$. The diagonal elements are the squared roots of the AVE.

*3.4. Hypotheses Testing*

The SEM is used to test the research hypothesis in this study. The model fitting parameters reveal that the absolute fit indices SRMR and RMSEA are 0.027 and 0.033, respectively, which are less than the standard of 0.05, whereas GFI and AGFI are 0.964 and 0.948, respectively, higher than the standard of 0.9. In addition, the value-added fit indices NNFI, NFI, CFI, IFI, and RFI are 0.984, 0.962, 0.987, 0.988, and 0.953, respectively, all are in the ideal value range of >0.9. Moreover, PGFI and PNFI are 0.675 and 0.770, respectively, higher than the standard of 0.5, and $\chi2/df$ is 1.475, also within the standard value range, suggesting that the data fitted well with the model. Figure 3 shows the test results of the model hypotheses. Tourists' perceived benefit directly impacts the environmental protection attitudes (H1: $\beta = 0.627$, $p < 0.001$) and resource utilization attitudes (H2: $\beta = 0.466$, $p < 0.001$), and positively impacts the consumption intentions of tourist (H3: $\beta = 0.427$, $p < 0.001$), thereby supporting H1, H2, and H3. Furthermore, the environmental protection attitude did not significantly impact the tourism consumption intention (H5: $\beta = 0.123$, $p = 0.091$), while the resource utilization attitude positively impacts the tourism consumption intention (H6: $\beta = 0.272$, $p < 0.001$). Hence, H5 is not supported, while H6 is supported.

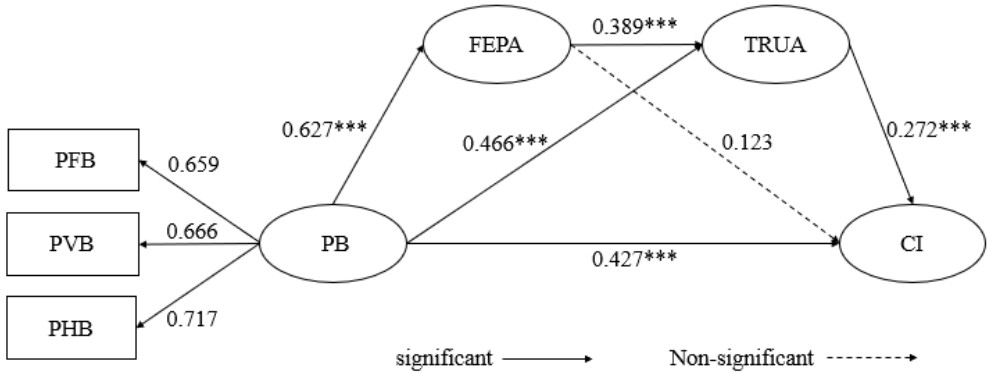

**Figure 3.** Model parameter estimation results. Note: *** $p < 0.001$. PFB = Perceived Functional Benefit; PVB = Perceived Value Benefit; PHB = Perceived Health Benefit.

*3.5. Multiple Mediation Test*

This study has used the deviation-corrected nonparametric percentile Bootstrap method for repeated sampling 5000 times to test the mediating effect; Table 3 shows the results of this method.

Firstly, the comprehensive effect of forest tourists' perceived benefit on their consumption intention is analyzed in order to determine whether a direct correlation exists between the two variables or whether their relationship is mediated by other variables. The test results reveal that the total effect 95% CI is [0.606, 0.775], the total indirect effect 95%CI is [0.155, 0.386], and the CI excluded 0. Moreover, the direct effect of 95% CI is [0.256, 0.606], that is, the total effect, indirect effect, and direct effect of tourists' perceived benefits on

their consumption intention are all significant. This confirms a partial mediating effect between perceived benefit and consumption intention.

**Table 3.** Mediating effect test.

| Effect | Path | Standardized Coefficient | 95% CI |
|---|---|---|---|
| Specific indirect effect | Ind 1: Perceived benefit → Environmental protection attitude → Consumption Intention | 0.077 | [−0.008, 0.161] |
| | Ind 2: Perceived benefit → Resource utilization attitude → Consumption Intention | 0.127 | [0.056, 0.221] |
| | Ind 3: Perceived benefit → Environmental protection attitude → tourism resources utilization attitude → Consumption Intention | 0.066 | [0.024, 0.132] |
| Total indirect effect | Perceived benefit → Consumption Intention | 0.271 | [0.155, 0.386] |
| Direct effect | Perceived benefit → Consumption Intention | 0.427 | [0.256, 0.606] |
| Total effect | Perceived benefit → Consumption Intention | 0.698 | [0.606, 0.775] |
| Specific indirect effect comparison | Diff 1 = Ind 3–Ind 2 | −0.061 | [−0.144, −0.010] |

Secondly, the multiple mediating effects of forest tourists' attitudes toward environmental protection and resource utilization are analyzed in this study. Taking tourists' perceived benefits as the independent variable, tourism consumption intention as the dependent variable, and attitudes toward environmental protection and resource utilization as continuous intermediaries, the model charted three pathways. The specific mediating effects for the three pathways are also analyzed, as shown in Table 3. (1) The path test results of SEM reveal that the attitude toward environmental protection exerts a positive impact on the intention of tourism consumption; however, this impact is statistically insignificant. Thus, the path of "perceived benefit → environmental protection attitude → consumption intention" (Ind 1) is not supported, suggesting that the independent mediating effect of environmental protection attitude is not significant. (2) In the path of "perceived benefit → resource utilization attitude → consumption intention" (Ind 2), the mediating effect of resource utilization attitude is 0.127 (95% CI [0.056, 0.221]), and the CI does not contain 0, i.e., the independent mediating effect of tourists' resource utilization attitude is significant. (3) The environmental protection attitude and resource utilization attitude are continuous chain mediators that create the path of "perceived benefit → environmental protection attitude → resource utilization attitude → consumption intention" (Ind 3). The mediating effect of the chain is 0.066 (95% CI [0.024, 0.132]), and CI does not include 0, suggesting that this effect is significant.

Thirdly, this study has also compared and analyzed specific mediating effects. The mediating effect (Ind 3–Ind 2) between paths Ind 3 and Ind 2 is −0.061 (95% CI [−0.144, −0.010]), and CI does not contain 0, suggesting that the mediating effects of these two paths are significantly different. Finally, the impact of the direct and mediation effects are compared in this study. The total mediating effect of the three mediating paths is 0.271, and the total mediating effect quantity is 38.83%, which is relatively small compared with the direct effect (effect value = 0.427; effect quantity = 61.17%), suggesting that the perceived benefits of forest tourism are the direct and primary influences on their tourist consumption intention.

## 4. Discussion

### 4.1. Theoretical Implications

Firstly, this study establishes that forest tourists' perceived benefits exert a direct positive influence on their consumption intention. The direct effect coefficient of tourists' perceived benefits on consumption intention is 0.427, accounting for 61.17% of the total effect coefficient of 0.698. The study results support previous studies that claim that the

perceived benefits of products or services are the main influencing factors of consumption intention [30,32]. Among the three perceived benefits (functional benefit, health benefit, and value benefit), functional benefits and health benefits have the highest mean values. This indicates that the tourists recognize the efficacy of forest tourism in psychological detoxification and physical strengthening, and they are eager to seek the satisfaction afforded by different types of experience, including recreation, medical care, and healthcare. This finding corroborates many studies' conclusions regarding the perceived benefits of forests [65,68,103]

According to the biophilia hypothesis, under the stimulation of natural environmental factors, people have natural connectivity [47], which leads to change in environmental attitudes [79]. This study draws upon the concept of "duality of environmental attitude" and differentiates between environmental protection attitude and resource utilization attitude. This study demonstrates not only that this distinction is meaningful, but also that the two environmental attitudes exert different mediating effects on the perceived benefit and consumption intention of forest tourists. Some studies highlight that tourists' perceived benefits in the natural environment could stimulate environmental protection attitudes [81,82,84,104]. However, few studies assess the correlation between them quantitatively. Through SEM, this study establishes that tourists' perceived benefit in forest tourism exerts a significant positive impact on their forest environmental protection attitude.

The attitude toward resource utilization is another dimension of environmental attitude, i.e., enjoying nature purely, using nature to realize personal entertainment, relaxation, and spiritual benefits [52]. Studies most often claim that the more benefits tourists perceive, the easier it is to create a positive attitude toward destinations and tourism activities [105]. Some other studies further investigated whether people would support the development of tourism resources due to perceived benefits [106]. However, such research primarily focuses on the local community residents [107] and seldom explore tourists' attitude toward resource utilization and its forming factors. In this regard, our study demonstrates that tourists' perceived benefit in forest tourism exerts a significant positive impact on the utilization attitude of forest resources.

This study also revealed that tourists' resource utilization attitudes exert a significant positive impact on consumption intention. However, tourists' attitude toward forest environmental protection exerts no significant impact on the consumption intention. Contrarily, many studies have reported that the stronger the tourists' attitude toward environmental protection, the more willing tourists are to pay for admission fees and environmental conservation fees [42,43]. There could be two possible reasons behind this. Firstly, tourists generally have sympathy for nature [108]. However, tourists who have attribution bias might not think that their tourism behaviors exert a negative impact on the environment. Thus, the impact of this protective mentality on behavior is unstable, and an attitude–behavior gap appears [109]. Secondly, tourists could lack a clear understanding of the ecological and environmental protection functions of forest tourism. Conversely, all these studies consider human tourism as acting in a contrary direction to nature conservation and environmental management programs [110].

The perceived benefits of forest tourists not only directly affect the consumption intention but also indirectly affect consumption intention through the attitude toward resource utilization. Zhao and An [95] discuss the correlation between health beliefs, attitudes, and behavioral intentions of Chinese forest tourists, and their results also reveal that tourists' attitudes toward nature play a mediating role between perceived health benefits and tourists' behavioral intentions. The study results support the SOR model and reveal a more complex causal relationship that has not been explored in previous studies: the positive effect of tourists' perceived benefits (S-stimulus) on willingness to consume (R-response) in forest tourism destinations is the result of multiple mediating effects of two organisms (attitudes towards forest environment protection and attitudes towards resource utilization). This study demonstrates that the satisfaction of forest tourists'

recreational, medical, health, and other benefits creates a positive development attitude toward the forest environment, thereby stimulating their consumption willingness.

Kaiser et al. [51] highlights that the attitude toward environmental protection and the attitude toward appreciation and utilization of nature influence each other. This study further validates the significant positive impact of the environmental protection attitude on the resource utilization attitude by exploring forest tourists. Moreover, the environmental protection attitude and resource utilization attitude constitute a chain intermediary between tourists' perceived benefit and consumption intention. Previous studies report that tourists are interested in protecting various biological entities (e.g., wild animals, vegetation, soil, and water quality) in natural tourism destinations and in supporting small-scale sustainable tourism centered on nature [96,111,112]. The contribution of this study is its argument that the correlation between environmental protection attitude and resource utilization attitude is clear through SEM. The natural environment experience exerts an educational effect on tourists, which can not only enhance tourists' environmental protection attitude but also further drive tourists' willingness to experience the natural environment and support sustainable tourism.

### 4.2. Managerial Implications

As people become more health conscious, ecotourism is becoming increasingly popular, and forest tourism destination management teams would do well to seize the current opportunities, particularly with the epidemic having a strong impact on global tourism. This study provides important insights into forest tourism destination management by extending the following points: (1) Perceived benefits are one of the fundamental factors that influence tourists' willingness to consume forest tourism. Tourism developers should actively utilize the forest ecological resources, forest landscape resources, forest cultural resources, and forest edible and medicinal resources of Siming Mountain to enrich tourists' recreational tourism product experiences and enhance their perceived benefits. However, considering the lowest dimensional perceived functional benefits score, managers should focus on enhancing the functional benefits of Siming Mountain, which is the highest elevation and largest forest park in Ningbo. While the recreational value of Siming Mountain has not been given enough attention, and although some attractions have been developed in the forest park, tourists may not consider its ornamental value to be high. Therefore, Siming Mountain needs to display its beautiful and unique natural landscape to tourists while maintaining its ecology, thus prompting the tourists to be more inclined not to wantonly destroy the local artificial attractions and, furthermore, stimulate the tourists to spontaneously protect the local natural forest environment, further making the tourists more willing to appreciate and experience the local tourism projects, which eventually leads to their intention to consume forest tourism products. A more sensible choice than relying on the so-called ticket economy for forest tourism in Siming Mountain is sustainable ecotourism. (2) Tourism developers should consider the environmental attitudes of tourists. Our study argues that the attitudes toward forest environment protection and attitudes toward resource utilization have intertwined effects on willingness to consume. In the process of tourism development, it is necessary to abandon the negative model of protection without development and to avoid the short-sighted approach of blind development. At the same time, forest biodiversity must be protected. Only one memorial hall has been built in Siming Mountain, but it is not associated with ecological education and therefore is not conducive to improving the environmental attitudes of the tourists. Managers would perform well to add environmental education facilities that serve both leisure functions in scenic areas while adhering to moderate development, such as setting up interesting information boards on the flora and fauna, establishing museums to exhibit specimens of local flora and fauna, and shaping the perception that consumption of forest tourism products can support the sustainable development of local scenic areas.

## 5. Conclusions

Based on the forest tourism consumption context, this study sub-categorizes the environmental attitude into environment protection attitude and tourism resources utilization attitude, and takes them equally as mediating variables to try to elucidate their mechanisms. Afterward, we examined the effects of forest tourists' perceived benefit (S—stimulus) and the two attitudes (O—organism) on consumption intention (R—response). The results revealed the following implications: (i) In the context of forest tourism, positive perceived benefits can directly promote tourists' intention to consume. (ii) Between the perceived benefits of forest tourism and tourists' consumption intention, resource utilization attitude can play a partial mediating role independently; that is, perceived benefits may affect tourists' consumption intention directly but also indirectly, through the intermediary role of the resource utilization attitude. (iii) Tourists' environmental protection attitude and resource utilization attitude play an intermediary chain-mediating role between tourists' perceived benefit and consumption intention, thereby forming the path of "forest tourists' perceived benefit → environmental protection attitude → resource utilization attitude → tourist consumption intention". This study enriches the research content of consumption intention in forest tourism. It provides a new perspective for understanding the psychological pathway of tourists participating in tourism activities with ecological functions.

## 6. Limitations and Future Research

Firstly, the data collection process for this study lasted two months. Future researchers should consider collecting samples from multiple time periods and expanding the amount and diversity of samples in order to longitudinally examine the changes in tourists' spending intentions during the development of the forest tourism destinations. Secondly, this study investigates the impact of perceived benefits of forest tourism on the consumption intention, rather than the actual consumption behavior of tourists. It should be noted that consumers' actual behavior is not always consistent with their declared behavioral intentions [109]. Thus, future research should focus on exploring the tourism consumption behavior of forest tourists. Thirdly, this is a cross-sectional study, and the impact of tourists' perceived benefits of forest health tourism on environmental attitudes could be overestimated. Before tourist activities, tourists could have formed different environmental attitudes. Therefore, in the future, the changes in the tourists' environmental attitudes from before and after their tourist activities should be examined in order to elucidate the difference, if any, in the impact of changes in environmental attitudes on tourists' consumption intentions. Finally, this study primarily considers the mediating role of environmental protection attitude and resource utilization attitude. Hence, future research should investigate the moderating role of consumers' green behavior and environmental ethics.

**Author Contributions:** Conceptualization, Methodology, Investigation, Writing—original draft, B.Z., S.L. and H.Y.; Project administration, Modify and editing, B.Z. and H.Y.; Supervision, Funding acquisition, Writing—review and editing, D.Z.; Data curation, Software, Writing—review and editing. S.L. and Q.X. All authors have read and agreed to the published version of the manuscript.

**Funding:** This study is supported by the National Natural Science Foundation of China (Grant No. 42171223), and Research on the Second Qinghai-Tibet Plateau Scientific Expedition (Grant No. 2019QZKK0401), Zhejiang Technical Institute of Economics research project (Grant No. JKY2021017).

**Data Availability Statement:** Not applicable.

**Acknowledgments:** Our great appreciation goes to the help from Siming Mountain National Forest Park management department.

**Conflicts of Interest:** The authors declare no conflict of interest.

## Appendix A

**Table A1.** The questionnaire was designed and used in this paper.

| Indicators | Items by Indicators |
|---|---|
| Perceived benefits (Defined as the outcome that a tourist perceives as conducive to meeting his expectations, motivations, and special needs while visiting a forest park) | **Perceived functional benefit**<br>The forest tourism of Siming Mountain has recreational value<br>The forest tourism of Siming Mountain has health preservation value<br>The forest tourism of Siming Mountain has medical value<br>The forest tourism of Siming Mountain has ecological value<br>The forest tourism of Siming Mountain has cultural value |
| | **Perceived value benefit**<br>Siming Mountain forest tourism has a distinct theme<br>Siming Mountain forest tourism has a reasonable consumption level<br>The price of Siming Mountain forest tourism is moderate<br>Siming Mountain forest tourism provides personalized services<br>Siming Mountain forest tourism provides professional services |
| | **Perceived health benefit**<br>Siming Mountain forest tourism can relieve pressure<br>Siming Mountain forest tourism makes people happy<br>Siming Mountain forest tourism can keep healthy<br>Siming Mountain forest tourism can enhance physical fitness<br>Siming Mountain forest tourism can take care of my body |
| Environmental protection attitude (Defined as tourist's attitude toward the practice of planting, maintaining, and protecting forest landscape for the purpose of conserving biological/natural and cultural values, sustainable use and equitable distribution of forest goods and services, and strategic preservation of forest resources for future use) | I think it's very important to protect the forest tourism resources and environment in Siming Mountain<br>Understanding the laws and regulations for the protection of recreational tourism resources in Siming Mountain<br>Understand the protection measures of forest recreation tourism resources in Siming Mountain<br>I'd like to participate in the forest tourism resources and environment protection in Siming Mountain<br>I'd like to prevent the destruction of forest tourism resources and the environment in Siming Mountain |
| Tourism resources utilization attitude (Defined as tourist's attitudes toward the appreciation of the destination environment and the use of its tourism resources) | It is necessary to develop forest tourism in Siming Mountain<br>The development of forest tourism in Siming Mountain should reflect local characteristics<br>The development of forest tourism in Siming Mountain should reflect the health-preserving culture<br>I would like to see more forest tourism and recreation projects in Siming Mountain<br>I would like to experience new forest recreation tourism projects in Siming Mountain |
| Consumption intention (Defined as tourists' intention to consume and pay for forest tourism projects and products) | I would like to pay for the forest recreation tourism projects in Siming Mountain<br>I would like to pay for the forest recreation tourism products in Siming Mountain<br>I would like to pay for experiencing the forest recreation culture of Siming Mountain<br>I would like to pay for receiving the forest recreation education in Siming Mountain |

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
