# Peer review of "Perceived Benefits and Forest Tourists Consumption Intention: Environmental Protection Attitude and Resource Utilization Attitude as Mediators"

_forests, doi:10.3390/f13050812_

Round 1

Reviewer 1 Report

Dear Authors, 

I have read your article carefully and submit my report below. Hope you will find it helpful. Best regards.

The article presents research in the field of tourism carried out with the use of advanced statistical methods often used in this scientific field. My general remark is that the article is dominated by statistical key issues and problems that may be related to tourism in general, while the forest element is treated very casually, as if reluctantly attached to an extensive methodology. The significance of this study for forests and forestry should be analysed in-depth.

I have included some more detail comments below. There are also some unfortune (in my opinion) phrases, which should be rephrased or explained, if they are not clear.

lines 36-38 “In addition, forest therapy (….) reduction in anxiety, positive emotional states, etc.” – the long sentence should be rephrased; the “etc.” at the end should be developed. If this is a paper about forest tourism, the forest aspect should be explained more broadly; this applies to this sentence as well as the entire text.

39-40 Why is Korea mentioned here? Further in the Introduction the other country is China. If the Introduction was planned to present a literature review and studies from various countries, the two are not enough. As the Authors mentioned, “the wellness tourism industry is booming worldwide” – not only in China and Korea. How about the forest tourism, how specific is this kind of tourism? How is it particularly different from other types of tourism?

The study aims  are defined in the lines 96-98: „to investigate the impact of forest tourist perceived benefits on consumption  behaviour, as well as to analyse the mediating role of tourists’ attitudes toward environmental protection and resource utilization” – what in fact does it mean in the human -forest relations? What are the „forest tourist perceived benefits” ? Can you list them? What kinds of consumption behaviour of man in a forest do you mean? How is defined the tourists “attitude toward environmental protection and resource utilization” particularly in forest? 

42-43 “Forests containing abundant natural landscape serve as the context of forest ecosystem services “– can you explain and rephrase?

49-50 “National forest parks are (…) an important position for popularizing nature knowledge” – should be reformulated.

51-52 “There are 3,548 forest parks at all levels in China” - what kind of levels? How many? Is it related to the division of the China system of nature conservation?

76 “Existing researches have proven that ….” – Are there any non-existing researches?

102-240 Materials and methods: I think the content of this chapter do not correspond to its title. Rather, the content is an overview of the literature, views, or the evolution of the point of view that led to the formulation of research hypotheses. In this section the Reader receives answers to some basic questions which I mentioned above. So, some significant parts of the section should be transferred to Introduction. The hypotheses should be listed in some separate section, maybe under the Research design.

242-254 Study area should be a separate section, equal to the Research design or Methods. The question is whether the research is carried out to study the area and the processes taking place there (humans thinking and behaviour), or vice versa: methodology prevails over a site that may be random?

The location map and map illustrating some natural characteristics would improve the chapter and help the Reader. Some more detailed description of the site would be welcome: why the area is so attractive for tourists, what particular tourist attractions are present there, tourist infrastructure,  some more information about the forest, types of the tree stands, wildlife; the forest coverage rate is not enough.

244  “altitude of 600900 m” – there must be some mistakes.

261-267 From the section Questionnaire design the Reader finally does not find out what exactly were the respondents asked about in the second part of the questionnaire; what were those “items” determined by the Authors? Maybe the most important ones can be listed or the questions can be attached as an appendix?

289 add “software by….”

The Results contain a lot of statistical results which were obtained by the methods not explained in the Methods section. It seems that elements of results and methodological details are mixed. The chapter Methods should contain some more detail explanation of the statistical methods that were used in the research, and why they were chosen; some references to those methods. In the Data processing, only the names were mentioned. This might be a problem for the Readers who are not experts in statistics.

299-300 “the average family income of tourists was mainly 68.8%”– what does this figure mean (68.9 %)? 68.9 % of what?

In the Discussion 396-471 the Authors related their findings to some previous research results clearly emphasizing what is their particular input to the knowledge of the subject. This is clear and interesting section.

474-475 “we utilized environmental psychology to sub-categorize (…)”  – add methods/elements of environmental psychology.

The conclusions seem largely applicable to tourism in general, no conclusions that specifically relate to the forest environment.

495-501 It seems that the implications i and ii are a bit too far-reaching in relation to the research carried out: postulating building forest cities, forest wellness bases, and developing all tourist services while taking care of the environment. These proposals are very general, especially the aspect of environmental protection, as the presented study did not contain any details related to the methods of environmental protection. The Reader does not know the current facilities, tourist infrastructure and conditions of  the natural environment, which is extremely important for further recommendations and directing the further development of tourism.

496-497 what kinds of forest sports? What type of cultural experience?

There are some typing errors in the text.

Author Response

Dear review experts:

We have made detailed modifications according to the review comments. Please read our revised draft and comment reply. Please see the attachment.

Thank you for your wonderful comments! 

Reviewer 2 Report

I commend the authors on putting together a well written manuscript on an important topic. The research is sound and provides interesting results that relate to improve in our understanding of perceived recreation benefits and environmental attitudes.

I suggest the authors make a few revisions before formal acceptance. I will detail my suggestions below.

  • Occasionally the authors refer to past social research as if they are describing definitive findings. In reality, most social phenomena have multiple ways to categorize and understand attitudes, values, and behaviors. I would suggest the authors qualify their language in places to not sound so definitive.

  • Related to the above, there are a few key environmental values researchers who should be acknowledged. Specifically, the authors refer to researchers who have categorized environmental attitudes and infer that those are the best and only categorizations. There are many researchers who have addressed this topic. So, in addition to ensuring their descriptions do not sound so definitive (see above), they should mention researchers like Kellert (nine environmental values) and Rolston who have written extensively on the topic. I’m sure the authors could come up with other environmental writers.

  • In addition to expanding on environmental value writers, the authors should address the old and well researched “outcomes focused management” framework that worked to better understand recreation benefits on natural settings. This began in the United States, but that does not mean it would not apply here. Specifically, look at research conducted by Bev Driver, Perry Brown, Namyun Kil, Taylor Stein, Chad Pierskalla, and Dorothy Anderson, among many others. In fact, the book “Benefits of Leisure” was published in 1990 based on the work by Driver and Brown, and researchers have explored this topic for decades.

  • Throughout the manuscript, the authors use terms that I am unfamiliar with. I marked terms that I believe require more explanation.

  • Related to the above, I was confused on what the authors meant by “resources” at the destination. This is important to better explain, and I believe the authors should provide examples of what they mean. Sometimes it comes across as if the visitors are hunting or extracting timber resources, which I do not think is what the authors are trying to tell us.

  • The authors do an excellent job of explaining past literature that informed their methods; however, they placed that literature review in the “Materials and Methods” section. I believe this would be better placed in a Literature Review section. It is still appropriate to keep the hypothesis in that section.

  • Under the “Data Processing” section, the authors concisely say they did reliability analysis, but they do not tell us that they used Chronbach’s Alpha. They should tell us this. Also, they say they used two factor analysis procedures, but do not tell us why. More explanation for these analyses would help this section.

  • The Stimulus–Organism–Response (SOR) is a good theory to drive this research, but the authors do not refer much to it in the Discussion. Since it is such an important part of this research, I would suggest the authors provide more discussion on how their results informed and improved this theory in terms of forest recreation. This discussion could come in the “Implications” section. So far, this section is correctly describes management and planning implications, but it could also refer to theoretical implications.

  • One problem with researchers who focus on looking or relationships (particularly with the SEM model) is that they ignore basic results – like averages. When I looked at Table 1, I was interested in what items came out high and low in terms of mean scores. I understand that this is not one of the hypotheses, but I do believe this could be an interesting result and help inform the entire paper.

  • I believe the authors should address some issues with their sample in “Limitations.” Specifically, they sampled visitors in a short amount of time (end of March to early May). I’m sure these visitors are unique and do not represent all visitors to the area. This is not a problem, but it should be acknowledged as a limitation. Also, I need more clarity on the response rate. The authors state that they had an 82.34% response rate in terms of visitors who accepted a questionnaire and returned it; however, how many visitors a refused to take a questionnaire? If that information was recorded, it should be explained.

Again, I commend the authors on providing a high quality manuscript on a good recreation topic.

Author Response

(The authors gave the same response as above.)

Reviewer 3 Report

This is an interesting study that looks at the interrelationships among variables such as public benefits, environmental protection attitudes, resource utilization attitudes, and consumption intention in a forest tourism area, with the two types of attitudes as chain-mediators between benefits and intention. The manuscript has merits for publication in forests, but the following concerns need to be addressed before its acceptance for publication in the journal. Specifically:

  1. for hypothesis 1, better to specify what type of consumptions are intended? WTP or others? You mentioned green consumption behaviors (line 147) while consumption is measured by WTP. So it would allow the reader to better follow the rest of the paper if consumptions are related to WTP in your hypothesis.
  2. need more info about the survey sites. First, reasons for choosing Siming Mountain National Forest Park as the study area. Second, I felt hard to do a random survey onsite even with a good plan like every 5th being picked as mentioned in this study for at least reasons: 1) not all can be stopped for the survey, then violating the rule of every 5th being chosen, 2) while it is easier to handle the survey on a trail or near a trailhead with people walking in a single file, it is hard to do so when you get people passing simultaneously (if the trail is wide), 3) what if you got a larger group more than 5 people? I would say it is more like a convenience survey than a random one in practice. Finally, were surveys carried out after a visitor completed their visit (or so-called exit survey)? (line 271-280).
  3. issues like data screening, missing data, etc. need to be reported.
  4. what is the rule to remove items with heavy cross loadings on two factors (line 307)
  5. Lines 199-211, it seems it is the use or appreciation of nature that stimulates the possible behaviors being environmentally friendly, thus it is appreciation that promotes or leads to protection. Along this line of thoughts, it is utilization that proceeds protection, or in other words, the latent variable on utilization attitudes seems more appropriate to be considered as an antecedent to the latent variable measuring protection attitudes. If this is true, the direction will be from TRUA to EPA (Figure 1).

Others

  1. Line 51, There are 3,548 forest parks at all levels in China, receiving more than 1.8 billion visitors. citation needed
  2. line 76, Existing researches have proven that tourists’ environmental attitudes comprise 76 individual values and beliefs about environmental protection and improvement (Joshi & 77 Rahman, 2019; Kaiser et al., 1999; Paço & Lavrador, 2017). Need to be aware of the conceptual hierarchy order of value, belief, attitude…They are not the same or attitudes are lower than values.
  3. line 244, with an altitude of 600900m, double check.
  4. line 267, the Likert seven-point scaleà the seven-point Likert scale

      5. lines 517-520, Finally, this study primarily considers the moderating role of environmental protection attitude and resource utilization attitude. Hence, future research should investigate the mediating role of consumers’ green behavior and environmental ethics. Which role considered in this study, moderating or mediating?

Author Response

Detailed listing of responses to the ID forests-1670414 entitled " Perceived benefits and forest tourists consumption intention: environmental protection
attitude and resource utilization attitude as mediator".
We are working on further language refinements until all requirements are met.
Thank you for your constructive comments and suggestions !

Round 2

Reviewer 1 Report

After the changes to the structure, and supplementation of the content, the text is clearer and more understandable, the improvements introduced by the Authors have organized the course of the reasoning, although some issues could still be described better (we still know little about the current tourism system and tourism infrastructure in the research area, although they are commented in the Discussion). The text in its current form is acceptable, after some minor corrections. I think some stylistic and editorial corrections are needed, especially in the final sections of the Discussion.

Line 261: is it an elevation in metres above sea level? Or any relative heights to some other reference level?

In Fig. 2 some elements are unclear, especially those on the right, in the map legend the river is marked with a blue polygon, which is associated with a surface object rather than a linear object, such as a river; it is difficult to distinguish rivers from the blue background meaning 'low elevation', what is the unit of the height range shown?

Line 273  - The section probably should start from the sentence that the research was based on the questionaries filled by forest tourists in Siming Mountain, and then continue: “ The first part of the questionnaire… “

It seems the questions mentioned in the lines 280-284, which are included in Appendix, are the same as the items in Table 1, so maybe it was enough to explain in the text that the cores of the most important questions included in the questionnaires were the same as those items presented in Table 1. I think the Editor may decide whether to include Appendix or not.

Author Response

Dear professor:

Detailed listing of responses to the ID forests-1670414 entitled " Perceived benefits and forest tourists consumption intention: environmental protection attitude and resource utilization attitude as mediators".

We invited a company called Meiji to help improve the language.

Thank you for your constructive comments and suggestions !

May 2022
